# Gamma Dose Rate Measurements in Northern Spain: Influence of Local Meteorological Scenarios on Radiological “False Alarms” in a Real-Time Radiological Monitoring Network

**DOI:** 10.3390/s24216812

**Published:** 2024-10-23

**Authors:** Natalia Alegría, Miguel Ángel Hernández-Ceballos, Igor Peñalva, Jose Miguel Muñoz

**Affiliations:** 1Energy Engineering Department, University of the Basque Country, 48013 Bilbao, Spain; igor.penalva@ehu.eus; 2Department of Physics, University of Cordoba, 14071 Córdoba, Spain; f92hecem@uco.es; 3Department of Industry, Basque Government, 01003 Vitoria, Spain; josemi-munoz@euskadi.eus

**Keywords:** ^222^radon, gamma dose rate, alarm level, Galerna

## Abstract

The present study characterizes gamma dose rate (GDR) concentrations in Bilbao, located in the northern Iberian Peninsula, utilizing a comprehensive 10-min interval database spanning from 2009 to 2018. This station belongs to the radiological environmental monitoring of the Basque Country network. The daily average GDR was found to be 0.07624 ± 0.00004 µSv/h, with the daily 95th percentile averaging 0.08026 ± 0.00007 µSv/h throughout the entire period. Our analysis specifically addresses the impact of precipitation on GDR, revealing a higher correlation coefficient for daily 95th percentile values compared to daily averages. Additionally, the influence of the Galerna (GL) event, a natural meteorological phenomenon in this region, on GDR was investigated, noting that it can develop both with and without precipitation. Understanding the interaction between GDR and this meteorological scenario is vital for the development of more reliable radiological monitoring strategies and for safeguarding public health. For this purpose, 40 GL events were analyzed. The present results indicate that GDR values frequently exceed alarm levels when a GL is formed. These GDR peaks should be considered natural radiological events, necessitating the classification of such GDR peaks as false alarms within the radiological monitoring network. To explain them in detail, 10-min time series of precipitation and radon outdoor concentrations were analyzed. The results demonstrate that the GL event with precipitation is a meteorological scenario that can be associated with false alarms. The present analysis provides a distinct contrast in radon behavior under the same meteorological event in case of being developed with precipitation or without precipitation. The findings from this analysis are crucial for avoiding and understanding false radiological alarms triggered in the monitoring network, thereby enhancing the accuracy of radiological data interpretation and improving public safety measures.

## 1. Introduction

The gamma dose rate (GDR) is a crucial parameter in radiation protection, representing the rate at which gamma radiation is absorbed by a medium. Gamma radiation, being highly penetrative, poses significant health risks, including increased cancer risk upon prolonged exposure. Therefore, monitoring and understanding the GDR is essential for public health and safety, environmental monitoring, and regulatory compliance [1,2].

Gamma radiation originates from various natural and artificial sources. One significant natural source is radon, a radioactive gas emanating from the decay of uranium and thorium in soil and rocks. Radon concentration in the atmosphere varies widely from place to place, with an average value in the atmosphere of 10 Bq m^−3^ [3,4]. Radon itself emits alpha particles; however, its decay products, such as lead-214 and bismuth-214, emit gamma radiation, contributing to the GDR in outdoor environments. Recent studies have highlighted the influence of outdoor radon concentrations on the GDR. For instance, a study conducted across several European countries established a correlation between elevated radon levels and increased GDR readings. In the Iberian Peninsula, similar studies have been conducted to understand the relationship between outdoor radon concentrations and GDR. For example, a study in central Spain found a direct correlation between radon concentrations and GDR, with higher levels observed in granitic regions. Another study emphasized the seasonal variability of radon and its impact on GDR and noted higher values in the summer months due to the increase in soil exhalation rates. These findings underscore the importance of continuous monitoring of both radon concentrations and GDR in order to enhance the understanding of their interplay and, hence, to inform better risk assessment models and mitigation strategies, ultimately safeguarding public health.

Meteorological conditions play a pivotal role in the measurement of GDR and outdoor radon concentrations. Temperature, humidity, atmospheric pressure, and precipitation are significant factors that influence radon exhalation from the ground and its subsequent contribution to GDR. For instance, higher temperatures can enhance radon exhalation rates from the soil, increasing atmospheric radon concentrations and, consequently, GDR, while wind direction and speed play a crucial role in the spatial and temporal distribution of radioactive particles, affecting both the intensity and distribution of GDR recorded at different locations [4]. In the case of precipitation, it is accompanied by an abnormally rapid increase of GDR, which is explained by the radon washout process [5].

In the context of radiological monitoring, the detection of elevated GDR at radiological monitoring stations necessitates a thorough analysis to distinguish between genuine radiological events and false alarms. Proper differentiation between true and false radiological threats is critical for effective radiological surveillance and public safety management. Elevated readings can be indicative of significant radiological incidents, but they may also result from non-hazardous sources such as natural radon progeny, cosmic radiation variations, or local environmental conditions. The background GDR varies mainly as a result of the deposition of ^222^Rn progeny in precipitation [6]. In this context, it is imperative to study and characterize the relationship between GDR, radon concentrations, and precipitation events. Recent studies have underscored these relationships, such as [7,8]. Understanding these natural influences is then crucial for improving the accuracy and reliability of radiological surveillance systems, thereby ensuring that genuine threats are promptly identified while minimizing unnecessary alerts and associated responses.

This article aims to further elucidate these relationships among GDR, radon concentrations, and the impact of precipitation, to provide a comprehensive analysis of the current understanding and its implications for radiation safety. To this purpose, GDR from 2009 to 2018 measured in Bilbao (northern Spain) is analyzed (1) to characterize GDR concentrations; (2) to establish the impact of precipitation on GDR concentrations; (3) to analyze the impact of the occurrence of “Galerna” (hereafter, GL) [9], a local meteorological event developed in this area, which can be developed both with and without precipitation; and (4) to establish correlations and dependencies between variations in GDR and ^222^Rn levels under GL events, with precipitation and without precipitation. Therefore, by analyzing data collected over this period the study intends to elucidate how fluctuations in GDR concentrations are influenced by precipitation and radon concentrations under the development of GL events.

Conducting this analysis, the results provide a significant benefit for the environmental radiological surveillance network of the Basque Country. By examining the correlations between radon levels and GDR during these events, the study helps refine the programmed alarm levels at the station. This refinement is crucial because exceeding certain thresholds during GL periods can indicate unnatural spikes in radiation levels. Understanding the typical behavior of gamma dose rates during this period, under dry and wet conditions, enables more accurate differentiation between natural and anomalous radiation levels, thereby enhancing the reliability and effectiveness of the station’s radiological monitoring efforts.

## 2. Materials and Methods

### 2.1. Study Area

The research area covers the Basque Country, located in the northern part of the Iberian Peninsula, bordered by the Pyrenees and the Cantabrian Mountains. This area features a varied topography with moderately low mountains, the highest of which range from 1000 to 1500 m above sea level, and narrow valleys that shape its terrain. Bilbao, the largest city in the Basque Country, with a population exceeding 340,000 inhabitants, is situated in the Nervión valley, approximately 16 km inland from the Bay of Biscay (Figure 1). Meteorologically, Bilbao experiences a temperate maritime climate strongly influenced by its proximity to the Atlantic Ocean. The city receives significant annual rainfall, averaging over 1000 mm, which is evenly distributed throughout the year, with relatively higher precipitation in the winter months. The prevailing wind directions in Bilbao are from the south and north [10], due to the channeling effect created by the surrounding mountains and valleys.

### 2.2. Data: Gamma Dose Rate, Radon, and Meteorological Observations

Ten-minute time series of GDR, radon, and meteorological observations, such as temperature (°C) and precipitation (mm), were used. To analyze the impact of precipitation on GDR levels, non-precipitation (NP) days were defined with a daily amount of precipitation equal to 0 mm, while precipitation (P) days were those with a daily amount of precipitation above 0.1 mm. In the automatic radiological surveillance network of the Autonomous Community of the Basque Country (Spain), one of the stations is located at the Bilbao School of Engineering. The Bilbao radiological station is equipped with a BAI9100, which measures alpha and beta particle concentrations, iodine, radon, and gamma dose rate (GDR). Alpha, beta, and radon are detected using a ZnS plastic scintillator; iodine is measured with a NaI detector; and GDR is monitored with a proportional counter. Briefly, radon is estimated on the basis of its progeny. The reader is referred to Hernández-Ceballos et al., 2023 [11] and the manual of BAI9100 to obtain more information about how the radon time series are obtained.

### 2.3. Definition of Alarm Level

To define alarm levels, it is essential to identify and understand fluctuations in GDR values. Using the recorded GDR values, two distinct alert thresholds are defined based on weather conditions: one for dry conditions, and another for wet conditions. The GDR values for both scenarios follow a normal distribution, and given that the data consist of discrete values, we apply Currie’s definition [12] to establish the critical levels of GDR. The alert threshold for dry weather is activated in the absence of precipitation, while the wet weather threshold becomes active during precipitation events. These thresholds adjust dynamically depending on the presence or absence of rainfall, ensuring that variations in GDR are properly assessed according to prevailing weather conditions. The methodology to calculate these threshold values uses GDR data recorded in the previous year, and two alarm levels are determined, one for dry and another for wet weather conditions, following
Lc=k⋅σ
where, *σ* is the standard deviation of the normal distribution fitted to the recorded GDR values for non-precipitation and precipitation days, respectively, and *k* is the coverage factor chosen to ensure the confidence level desired.

These alarm levels are calculated for each station and automatically stored for real-time identification. The alarm levels are then compared with GDR values every 10 min. The procedure is the following: Each time that a GDR measurement is recorded, it is checked whether there is a record of precipitation or not. Based on this, the recorded value is compared to the corresponding alarm level. If the GDR value exceeds the alarm level, an automatic notification is triggered. Proportional counters are affected by storms, fog, voltage fluctuations, nearby construction activities, and other factors. For this reason, this procedure must be well-defined, as setting the alarm threshold too high could allow critical alerts to be missed, while setting it too low would trigger continuous alerts, diminishing their significance. The values of alarm levels for dry and wet conditions during the period 2009–2019 are shown in Table 1.

### 2.4. Galerna Events

The set of GL events from 2009 to 2018 identified by Gangoiti et al., 2023 [9] has been taken as a reference. Basically, GL is a sudden strong or very strong and gusty wind, accompanied or not by precipitation, typical of the northern coast of Spain [9], characterized by the occurrence and development of this event, indicating that it occurs more frequently between May and June, and between noon and the late afternoon. More information about GL can be consulted in [9]. It is necessary to indicate that there were no GL events in January and December for the period analyzed. Data quality control was implemented due to occasional data loss or downtime at the station. For a GL event to be considered valid in the present study, a threshold of 75% availability of 10-min GDR data was established. Therefore, GL days with lower data availability were excluded from the analysis. Applying this control, there were no GL events in July, and the total number of GL events analyzed was 40 during the period 2009–2018. A total of 22 out of 41 were GL accompanied with precipitation (P) during the day, while 18 were GL without precipitation (NP) during the event. Figure 2 displays the monthly distribution of these 40 GL events during the period 2009–2018. Days with precipitation are noted in almost every month, with the highest number recorded in May (4 days).

## 3. Results and Discussion

### 3.1. Gamma Dose Rate in BILBAO (2009–2018)

The temporal evolution of daily averages and the 95th percentile of GDR at Bilbao city during the study period is shown in Figure 3, as well as the total amount of precipitation registered on each day. The average daily GDR concentration is 0.07624 ± 0.00004 µSv/h, where the uncertainty is given as the standard error of the mean. This value quantifies how much the mean is expected to vary due to random sampling errors, and a smaller value indicates that the sample mean is a more precise estimate of the population mean. This value decreases as the sample size increases, showing greater confidence in the mean estimate with larger samples. Daily GDR values reach peaks up to 0.089 µSv/h. The average GDR recorded in this city can be compared to values obtained across Europe. An example of this can be consulted through the radiological monitoring data collected in almost real-time by the European Radiological Data Exchange Platform (EURDEP, [13]). The 95th percentile values show higher variability, indicating occasional spikes in gamma radiation levels. The maximum value is 0.14 µSv/h, with an average value of 0.08029 ± 0.00007 µSv/h during the whole period. The precipitation data show significant variability, with some days experiencing heavy rainfall, with values above 60 mm.

The correlation values obtained between the daily mean values and the 95th percentile with precipitation show an increase, from 0.27 for daily mean values to 0.34 for P95 daily values, suggesting that precipitation is one of the primary drivers of reaching high GDR levels at this location. In this sense, the average during the whole period 2009–2018 of daily and P95 GDR concentrations are higher under precipitation (0.07655 ± 0.00001 µSv/h and 0.0811 ± 0.0002 µSv/h, respectively) than without precipitation (0.0759 ± 0.0008 µSv/h and 0.07955 ± 0.0008 µSv/h, respectively).

The seasonal GDR values with and without precipitation show higher concentrations during precipitation events across all seasons. In both meteorological scenarios, the highest average values are reached in winter (0.0769 ± 0.0002 µSv and 0.0765 ± 0.0001 µSv/h, for precipitation and without precipitation, respectively), while the lowest values are observed in summer (0.0754 ± 0.0002 µSv and 0.0751 ± 0.0001 µSv/h, respectively). Figure 4 displays the frequency distribution for GDR, showing a distribution centered between 0.076 and 0.078 µSv/h. Figure 4b shows the seasonal distribution of GDR values in Bilbao. Despite the similarity in the Gaussian shape of the distribution, differences are observed between the distributions depending on the season in terms of the peak frequency occurrence. While in the winter, spring, and autumn the peak is reached in the range of 0.076 to 0.078 µSv/h, in the summer the peak frequency is reached in the range of 0.074 µSv/h to 0.076 µSv/h. This implies a steeper positive slope in the summer curve compared to the other seasons, which do not exhibit such an abrupt rise. Conversely, these seasons show higher values than those observed in summer. During the summer, lower GDR values are typically observed, primarily due to reduced radon exhalation from the soil and enhanced atmospheric dispersion driven by warmer temperatures and increased solar radiation.

Finally, the influence of precipitation days on the GDR values is shown in Figure 5, which displays histograms for each season, comparing the GDR concentrations on days with precipitation (P) and without precipitation (NP). Seasonal variations in GDR concentrations are observed, with higher values typically recorded in autumn and winter compared to summer [14]. P and NP days present the maximum frequency in the range “0.076–0.078” µSv/h in winter, spring, and autumn, while this peak is reached in summer in a lower range “0.074–0.076”. NP days generally have higher frequencies of lower GDR concentrations across all seasons, while, on the contrary, P days exhibit a higher frequency in the highest values, presenting a broader distribution of GDR concentrations. These results demonstrate the significant influence of precipitation on GDR concentrations, primarily due to the washout effect, where rain brings airborne radionuclides closer to the ground, leading to increased GDR levels during and shortly after rainfall events [15]. Additionally, the accumulation of water on the ground can slightly reduce radon outgassing, tempering the increase in GDR. This seasonal pattern can be attributed to several factors, including increased radon exhalation from the soil during colder months and the accumulation of radionuclides on the ground surface, which are then washed out by precipitation [10].

### 3.2. Gamma Dose Rate Under GL Events, and Influence of Precipitation

In this section, 40 days with GL events from the period 2009 to 2018 are analyzed, all of which have sufficient hourly GDR data available. Figure 6 shows the overall average daily GDR cycle in Bilbao without GL (NGL) and the GDR cycle under GL events (GL). On average, GDR values under GL are higher throughout the entire day. It is notable that both cycles exhibit high values during the night, followed by a decline to lower values. However, from midday onwards, GDR concentrations rise under GL events, with an increase from 12:00 h to 20:00 h, after which they begin to decrease. This increase coincides with the period established in [9] for the development of GL. Therefore, the GDR cycles in Bilbao reveal distinct daily patterns, suggesting a positive influence of GL formation in GDR levels during the afternoon, emphasizing the impact of atmospheric conditions on radiation levels throughout the day.

GL events can be developed with and without precipitation [9]. Figure 7 displays the set of 40 days of GL covering the period from 2009 to 2018, indicating the total precipitation recorded for each one, as well as the daily average value of GDR. On days with precipitation, the average of GDR is higher (0.0781 ± 0.0005 µSv/h) than the average value on days without precipitation (0.0763 ± 0.0005 µSv/h). The average value ranges from 0.0733 µSv/h to 0.0828 µSv/h in precipitation days, while it ranges from 0.0720 µSv/h to 0.0790 µSv/h in non-precipitation days.

Figure 8 displays the hourly evolution of GDR under GL events, separating events with precipitation and without precipitation. Both daily cycles are compared with the overall average of GDR under GL events for the period 2009–2018. The figure shows that during GL events with precipitation (P), daily GDR concentrations are higher than those during GL days without precipitation (NP). Similar values for P and NP events are registered at night and in the morning. While those obtained for NP days present similar values to the overall average, GDR for P days is slightly higher. Notably, after 12:00 h, coinciding with the period in which [9] indicates the onset of GL development, the GDR on P days (red line) consistently exceeds both NP days (blue line) and the overall average (black line). This indicates a significant increase in the afternoon in GDR levels during GL events under precipitation, suggesting that precipitation may enhance the washout of radionuclides from the atmosphere, leading to elevated GDR levels. This evolution highlights that precipitation amplifies the effects of GL on GDR concentrations, particularly during critical periods of GL formation.

### 3.3. Impact of Radon Concentrations on GDR

To investigate the relationship between GDR and radon under GL events, Figure 9 shows the hourly evolution of radon concentrations under GL events, separating events with precipitation and without precipitation. In both cases, radon concentrations present the expected high concentrations at night and early morning due to the stable atmospheric conditions and low wind speeds, which reduce the dispersion of radon. However, radon levels show a small increase in concentration during the afternoon. This radon behavior during GL periods has already been observed in [11]. Within this general pattern, radon concentrations differ from the average daily cycle, depending on whether GL occurs with or without precipitation. In the case of GL with precipitation (Figure 9), radon concentrations are systematically lower than the average cycle, showing a tendency toward lower concentrations that coincides with an increase in GDR during the afternoon (Figure 8). The largest differences from the mean are observed at the end of the day, with values of 1.7 Bq/m^3^. However, on days with GL without precipitation (Figure 9), radon concentrations are higher than the mean, with an increasing divergence from midday onwards, reaching a maximum at the end of the day, with differences at about 2.3 Bq/m^3^. This upward trend in radon concentrations coincides with a slight decrease in GDR levels during the afternoon (Figure 8).

The relationship between GDR and radon concentration during precipitation and non-precipitation days can be explained by several environmental factors. The differences in these evolutions can be attributed to the washout effect during precipitation, where rain transports radon progeny from the atmosphere, leading to increased GDR levels due to the deposition of gamma-emitting radionuclides. The contribution to the GDR in the air from ^222^Rn and its progenies is small (0.5 (nSv/h)/(Bq/m^3^, for EEC concentration, equilibrium factor is 0.5), but after precipitation events when significant amounts of progeny are deposited on the ground, significant temporary rises of gamma dose rate occur that could easily double or triple the background radiation dose during an hour or more [16,17,18,19,20,21]. During NP periods, the absence of these effects results in higher concentrations of radon in the air, i.e., radon tends to accumulate in the atmosphere due to a lack of precipitation to remove it, and it can be transported through the atmosphere after exhalation from the soil, and hence, with less influence on GDR concentrations. These observations are consistent with studies indicating that meteorological conditions, such as precipitation, significantly influence radon and GDR gamma dose rate variations during GL events.

We can also include in the analysis of the relationship between GDR and radon the different modalities of measurements. The way the radon system works is that air is drawn in from outside and flows over a ZnS plastic scintillator that counts alpha and beta particles. Gamma rays might be measured more consistently at the bottom, from all directions, in a homogeneous manner. This suggests that the radon counter detects only the radon in the air, which can indeed decrease during rain events because the radon is “forced” to remain very close to the ground, on the ground itself, or even below it. In this sense, during normal conditions (NP), radon follows a daily cycle of emanation, which can be altered by precipitation. Alegría et al. [22] present the impact of different factors, such as the measure methodology, in the correlation between GDR and radon concentration.

### 3.4. GDR Evolution Under GL with Precipitation and Non Precipitation: Study Cases

Two study cases of the GDR evolution under the GL situation are represented in this section, the first one under dry conditions (31 August 2009) (Figure 10a), and the second one under wet conditions (1 August 2009) (Figure 10b). The daily evolution of temperature, radon concentration, and GDR is illustrated in figures that capture data every 10 min. The red horizontal line represents the alarm level for GDR under dry or wet conditions, respectively. This line, in agreement with Section 2.4, presents different values under precipitation and without precipitation.

Figure 10a shows a significant event, with a sharp temperature drop of about 10 °C at 15:10 h, corresponding to the arrival of a GL to Bilbao, as noted by [11]. This event triggers a radon concentration spike between 16:30 and 20:00, peaking at 20 Bq/m^3^. Studies conducted in Europe and other regions have shown that natural events, such as the arrival of specific air masses or weather fronts, can lead to significant short-term increases in radon concentrations [23]. During this increase in radon concentrations, GDR values remain mostly constant but over time increase slightly; some GDR values eventually reach the alarm level. These occurrences should be classified as false alarms. However, it is not possible to establish a direct correlation of these false alarms with radon concentration, as the sudden increase in surface winds during the GL [9] affects both rand and GDR concentrations independently.

Figure 10b illustrates the influence of precipitation on GDR and radon concentrations under GL conditions in Bilbao. Precipitation began at noon and continued into the afternoon, with 10-min intervals recording values between 0.1 to 0.9 mm. The most significant precipitation peak occurred between 12:00 and 13:00, coinciding with a temperature drop and a rapid increase in GDR from 0.08 to 0.095 µSv/h, along with a decrease in radon concentrations from 23 to 19 Bq/m^3^. The GDR value exceeded the alarm threshold during precipitation, indicating a false alarm. A similar pattern was observed between 17:00 and 19:00 h, where the reduction in radon concentrations due to precipitation resulted in an increase in GDR, although it did not surpass the alarm level.

The observed phenomena can be explained by the interplay between precipitation, radon concentrations, and GDR. Rainfall can wash radon progeny from the atmosphere, temporarily increasing radon concentrations near the ground. This displacement process often results in increased gamma radiation detected at the surface due to the decay of radon progeny that settles on the ground. This mechanism is well-documented in various meteorological and radiological studies, which have reported similar patterns of reduced airborne radon and increased GDR during and after precipitation events [24].

## 4. Conclusions

The present analysis has characterized GDR concentrations in the city of Bilbao (northern Iberian Peninsula) using a long-term 10 min database from 2009 to 2018. The daily average during the whole period is 0.07624 ± 0.00004 µSv/h, while the average of the daily 95th percentile was 0.08029 ± 0.00007 µSv/h. The impact of precipitation is addressed, obtaining a higher correlation coefficient for the daily 95th percentile than for the daily values. The impact of a GL event, a natural meteorological phenomenon developed in this area, on GDR has also been analyzed. For this purpose, radon outdoor concentrations and precipitation data were used. Differences in GDR values have been observed between GL with precipitation and without precipitation. The analysis has shown increases in GDR under precipitation accompanied by reductions in radon concentrations when the GL is developed. These GDR concentrations should be recognized as natural radiological events, necessitating the classification of such GDR peaks as false alarms in the radiological monitoring network. Understanding these interactions enhances our ability to accurately interpret radiological data and improve public safety measures.

## Figures and Tables

**Figure 1 sensors-24-06812-f001:**
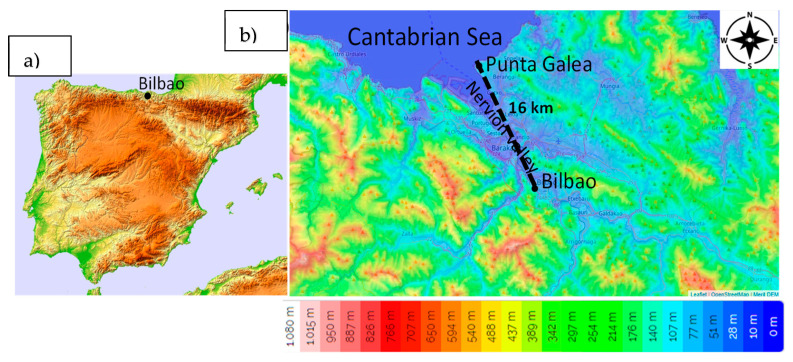
(**a**) Location of Bilbao in the Iberian Peninsula and (**b**) the topographic map of the surroundings of Bilbao.

**Figure 2 sensors-24-06812-f002:**
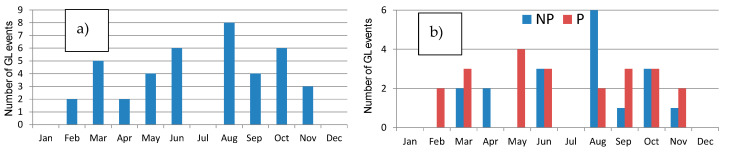
(**a**) Total number of GL events and (**b**) GL with precipitation (P) and non-precipitation (NP) in each month during the period 2009–2018.

**Figure 3 sensors-24-06812-f003:**
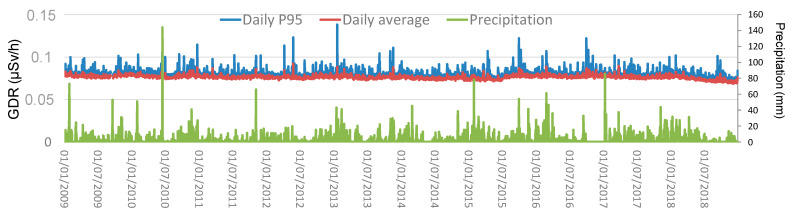
Evolution of daily averages and P95 values, and precipitation during 2009–2018 in Bilbao.

**Figure 4 sensors-24-06812-f004:**
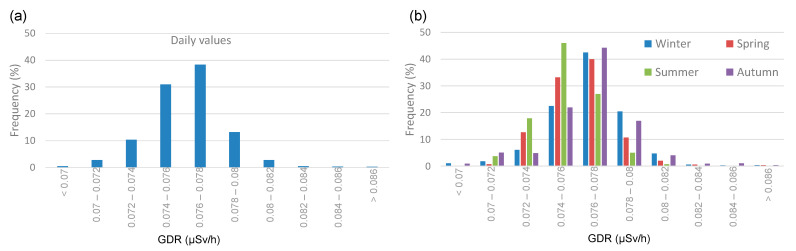
(**a**) Total and (**b**) seasonal histogram of GDR daily values during 2009–2018 in Bilbao.

**Figure 5 sensors-24-06812-f005:**
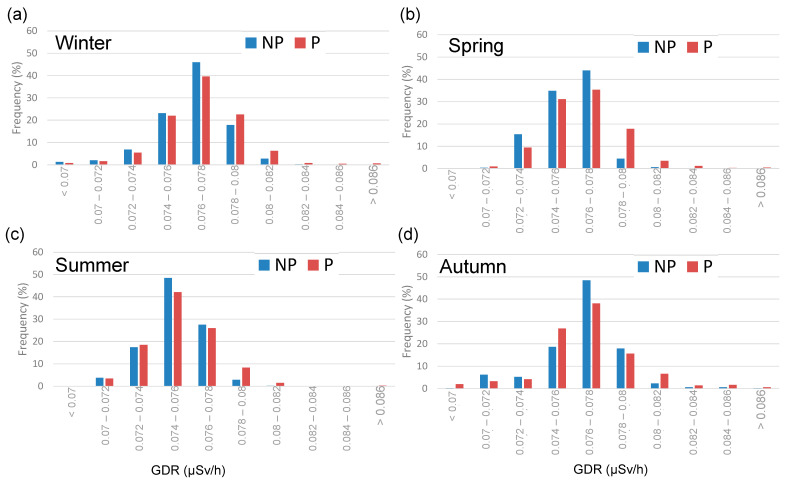
Seasonal histogram of GDR daily values during non-precipitation (NP) and pre-cipitation (P) days during 2009–2018 in Bilbao, (**a**) Winter, (**b**) Spring, (**c**) Summer and (**d**) Autumn.

**Figure 6 sensors-24-06812-f006:**
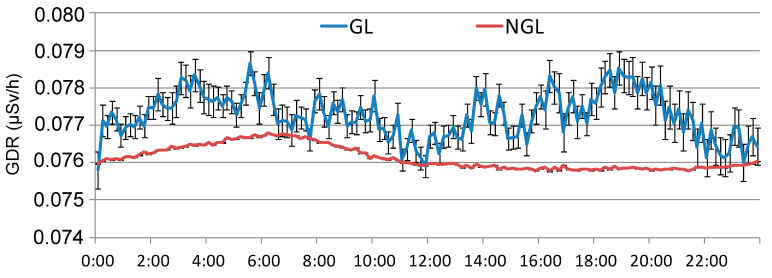
Daily cycle of GDR in Bilbao during the period 2009–2018 and under GL events during the same period.

**Figure 7 sensors-24-06812-f007:**
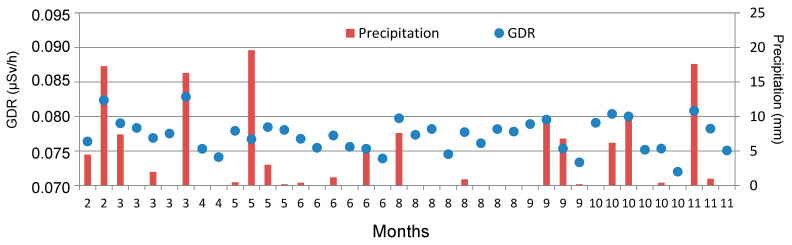
Amount of precipitation and GDR daily values of GL days in Bilbao during the 2009–2018 period. Note: GL events are ordered by months, from February to November. There are no GL events in January, July, and December.

**Figure 8 sensors-24-06812-f008:**
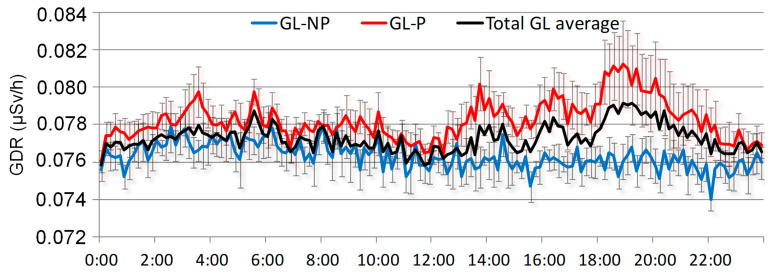
Daily cycles of GDR under GL events with precipitation (GL-P), non-precipitation (GL-NP), and the overall average during the period 2009–2018.

**Figure 9 sensors-24-06812-f009:**
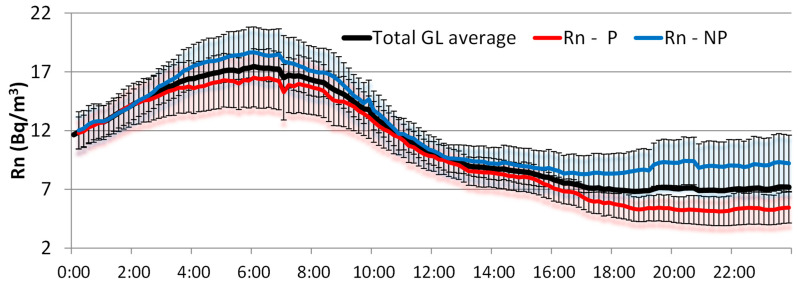
Daily cycles of radon concentrations under GL events with precipitation (GL-P), non-precipitation (GL-NP), and the overall average during the period 2009–2018.

**Figure 10 sensors-24-06812-f010:**
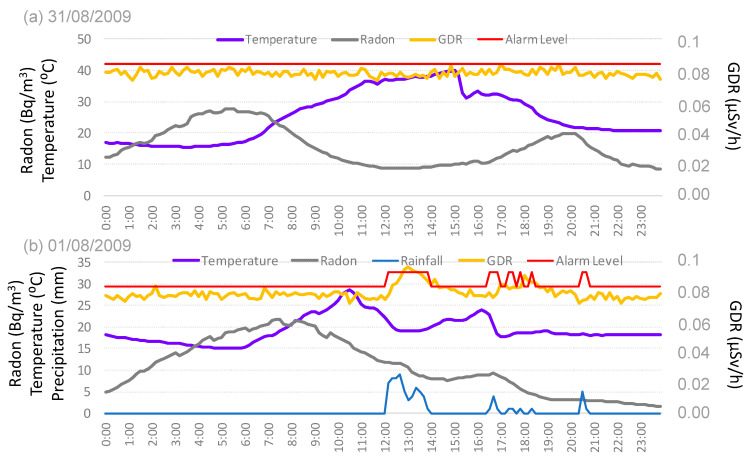
Daily evolution of GDR and radon concentrations, precipitation, and temperature under GL (**a**) without precipitation (31 August 2009) and (**b**) with precipitation (1 August 2009) in Bilbao. The red line represents the alarm level.

**Table 1 sensors-24-06812-t001:** Dry and wet alarm levels during the period 2009–2014 in the radiological station of Bilbao.

*YEAR*	Dry Conditions (µSv/h)	Wet Conditions (µSv/h)
*2009*	0.0841	0.0941
*2010*	0.0839	0.0930
*2011*	0.0830	0.0920
*2012*	0.0825	0.0914
*2013*	0.0828	0.0913
*2014*	0.0815	0.0900
*2015*	0.0839	0.0949
*2016*	0.0838	0.0913
*2017*	0.0829	0.0927
*2018*	0.0825	0.0904

## Data Availability

The original contributions presented in the study are included in the article, further inquiries can be directed to the corresponding author.

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
