# Peer review of "Gamma Dose Rate Measurements in Northern Spain: Influence of Local Meteorological Scenarios on Radiological “False Alarms” in a Real-Time Radiological Monitoring Network"

_sensors, 2024, doi:10.3390/s24216812_

Round 1

Reviewer 1 Report

Comments and Suggestions for Authors

This article analyzed the gamma  dose rate(GDR) database from 2009 to 2018 in Bilbao, presented the GDR level, the influence of precipitation on GDR. And the local environmental phenomenon, Galerna event influence to GDR was also discussed. In order to understand the influence, the radon concentrations with GL events w/o precipitation are analyzed, which find out that CL events with precipitation are associated with higher increased in GDR. This finding can help people understand the false radiological alarm, and then avoid to trigger fake alarms. The analysis result enhances the accuracy of radaological data which improve public safety service.

The data source and analysis method are described, then the results are given, including the GDR evolution under months, seasons, temperature, precipitation and local Galerna events. The shown data is adequate for the results.

It's excellent work and a good article.

Author Response

Review1: This article analyzed the gamma  dose rate(GDR) database from 2009 to 2018 in Bilbao, presented the GDR level, the influence of precipitation on GDR. And the local environmental phenomenon, Galerna event influence to GDR was also discussed. In order to understand the influence, the radon concentrations with GL events w/o precipitation are analyzed, which find out that CL events with precipitation are associated with higher increased in GDR. This finding can help people understand the false radiological alarm, and then avoid to trigger fake alarms. The analysis result enhances the accuracy of radaological data which improve public safety service.

The data source and analysis method are described, then the results are given, including the GDR evolution under months, seasons, temperature, precipitation and local Galerna events. The shown data is adequate for the results.

It's excellent work and a good article.

Many thanks for these positive words.

Reviewer 2 Report

Comments and Suggestions for Authors

Overall this is a very interesting study of a large set of measurements from a very special location that experiences a unique meteorological event. It is clear from this work that both precipitation and GL impact ambient radon concentrations and therefore GDR. However, it is not clear to me from this study if it was intended to show any significant difference between GL with and without precipitation.

It appears that both precipitation and GL impact GDR, but that these impacts do not necessarily compound with any significance. I'd like to see a more clear description of this in the conclusions along with more details regarding the determination of uncertainty in the presented average values. I find it very hard to believe that the standard deviation of the GDR reported over this period is only 0.05% (0.00004 / 0.07624 per the manuscript)... this may be due to my relative ignorance in this specific source of radiation but seems highly unlikely.

In a similar vein, the conclusions discussed in section 3.2 seem to ignore the overlapping measurement results. The values of 0.0781 and 0.077 are not sufficiently separated when considering the measurement uncertainty of 0.008. The conclusion that these values are different should be qualified with the confidence interval. 

These comparisons will be more obvious if uncertainties were included on most of the figures in this manuscript. I recognize that can be challenging with this amount of data... I would recommend including error-shaded regions in most figures as opposed to error-bars which would clutter the plots. This is particularly important for Figure 7.

Figure 11 refers to GDR evolution under GL with precipitation but the prior paragraph which refers to figure 11 does not discuss GL at all. Either this is an error with the caption or I mis-understood the objective of this section.

I will request additional citation for several statements I found lacking in the manuscript:

1) P.2 ln 50 and 52 - "higher levels observed in granitic regions" & "increased soil exhalation rates"

2) P.5 ln 198 

3) P.8 ln 257 - "lowering its concentrations during the afternoon"

Author Response

Review2: Overall this is a very interesting study of a large set of measurements from a very special location that experiences a unique meteorological event. It is clear from this work that both precipitation and GL impact ambient radon concentrations and therefore GDR. However, it is not clear to me from this study if it was intended to show any significant difference between GL with and without precipitation.

Many thanks for your words. About the last comment, we have included the following information in the new version of this article

  • Lines 17-18: Additionally, the influence of the Galerna (GL) event, a natural meteorological phenomenon in this region, on GDR was investigated, noting that it can develop both with and without precipitation.
  • Lines 82-89: 3) to establish the impact of precipitation on GDR concentrations, 3) to analyse the impact of the occurrence of “Galerna” (hereafter, GL) [9], a local meteorological event developed in this area, which can be developed both with and without precipitation, and 4) to establish correlations and dependencies between variations in GDR and 222Rn levels under GL events, with precipitation and without precipitation. Therefore, by analyzing data collected over this period the study intends to elucidate how fluctuations in GDR concentrations are influenced by precipitation and radon concentrations under the de-velopment of GL events.

It appears that both precipitation and GL impact GDR, but that these impacts do not necessarily compound with any significance. I'd like to see a more clear description of this in the conclusions along with more details regarding the determination of uncertainty in the presented average values. I find it very hard to believe that the standard deviation of the GDR reported over this period is only 0.05% (0.00004 / 0.07624 per the manuscript)... this may be due to my relative ignorance in this specific source of radiation but seems highly unlikely.

Many thanks for this comment/suggestion. We have tried to solve it in this new version of this article. In this sense it is necessary to remember that we calculate the standard error of the  mean, and not the standard deviation of the mean. We have included the following information in the new version of this article:

  • Lines 187-193: The average of daily GDR concentration is 0,07624 ± 0,00004 µSv/h where the uncertainty is given as the standard error of the mean, which is calculated by σ/N1/2, being σ he standard deviation of the sample and N the number of values in the sample. This value quantifies how much the mean is expected to vary due to random sampling errors, and a smaller value indicates that the sample mean is a more precise estimate of the population mean. This value decreases as the sample size increases, showing greater confidence in the mean estimate with larger samples.

In this line, we have modified several figures in the new version of this article, including the uncertainty of the mean values:

Figure 6

Figure 8:

In a similar vein, the conclusions discussed in section 3.2 seem to ignore the overlapping measurement results. The values of 0.0781 and 0.077 are not sufficiently separated when considering the measurement uncertainty of 0.008. The conclusion that these values are different should be qualified with the confidence interval. 

Many thanks for this comment/suggestion. There was a mistake in the calculated values. We have modified the information provided in the new version of this manuscript: Lines 270-274 “It is calculated that on days with precipitation, the average of GDR is higher (0,0781 ± 0,0005 µSv/h) than the average value on days without precipitation (0,0763 ± 0,0005 µSv/h).”

These comparisons will be more obvious if uncertainties were included on most of the figures in this manuscript. I recognize that can be challenging with this amount of data... I would recommend including error-shaded regions in most figures as opposed to error-bars which would clutter the plots. This is particularly important for Figure 7.

Many thanks for this comment/suggestion. We have included the error-bars in this new version of the manuscript, as we have previously indicated in this document. Figure 7 is one of the figures modified in the new version.

Figure 11 refers to GDR evolution under GL with precipitation but the prior paragraph which refers to figure 11 does not discuss GL at all. Either this is an error with the caption or I mis-understood the objective of this section.

Many thanks for this comment/suggestion. Yes, there was an error in the caption in the old version, and we have changed it in the new version of this manuscript.

I will request additional citation for several statements I found lacking in the manuscript:

1) P.2 ln 50 and 52 - "higher levels observed in granitic regions" & "increased soil exhalation rates"

It has been solved in the new version of this manuscript Lines 48-51 “For example, a study in central Spain found a direct correlation between radon concentrations and GDR, with higher levels observed in granitic regions, while another study emphasized the seasonal variability of radon and its impact on GDR, it noted higher values in the summer months due to the increase in soil exhalation rates.”

2) P.5 ln 198 

The paragraph has been changed in the new version of this paper.

3) P.8 ln 257 - "lowering its concentrations during the afternoon"

Many thanks for this suggestion. It has been changed in the new version of this manuscript.

Reviewer 3 Report

Comments and Suggestions for Authors

Overview:

The authors discuss measurements observed in a network of sensors deployed in the North Eastern region of Spain, around the city of Balboa. In particular, they discuss how a weather event in the region, called Galerna (GL), typically characterized by strong winds, and precipitation affects readings of the Gamma Dose Rate (GDR) in their sensors. They divide their dataset into precipitation (P) and non precipitation (NP) data, as well as GL and non GL data and investigate differences between these segments. Supplementary they also investigate radon readings and look for correlations with the GDR. Finally they present two examples of GL events, one with precipitation and one without and show how the discussed effect, are realized in individual instances.

Overall, the authors present their work in a clear and simple-to-understand language, the paper is well structured, and they use proper English. Their discussion of existing literature and their overall results agrees with the understanding that I have of the topic. The presented work is interesting and is relevant to the readers of sensors. I do agree with the content of the paper and the overall conclusion, but I don’t believe that most of the data, as they are currently presented, actually support their claims. In order to study the impact of events that typically last a few hours and can occur at any time throughout the day, it is necessary to decouple the daily fluctuations from the event's impact. This has not been done here. I believe most observed effects, as presented, are statistical in nature and just happen to go the right way (and sometimes they also go the wrong way)! I believe that it is possible to correct the analysis (and figures) and find quantitative indicative results. I will provide suggestions on how to go about this for each of the relevant sections in the general comments section below. If for some reason the suggestions do not work out, or if the authors already tried this, they should talk about it in a “Discussions” section. To conclude, I strongly encourage the authors to address the flaws of the manuscript and continue pursuing a publication in this journal. I believe their work is interesting and having more results in this field of research is very valuable.

One final comment: There are a lot of small errors (wrong Figure and Section numbers, for example) in the script. A proper proofreading should have caught those. It is not a reviewer's job to proofread a paper, the authors should have done so themself.

General comments:

Section 2.3: I don’t think it is entirely clear that you have two alarm levels. One being active whenever there is rain, another (lower one) whenever there is no rain. Try to make this a bit clearer. Also, I think the discussion of the alarm level here is cryptic. Maybe just give some examples and provide the false alarm rate. At k=1 you will have about 15.8% of backgrounds triggering an alarm (or roughly one false alarm every hour), at k=2 you have about 2.3% of backgrounds doing the same (or roughly a false alarm every 7 hours), k=3 about 0.1% of backgrounds (false alarm rate of 1 every 5 days). Please check my math. I personally would just state the false alarm rate for a few examples of k.

Section 2.4: What is the average length of a GL event? Would be good to mention this. Is it a few minutes, hours, or days? Is the GL event starting randomly or is there a correlation with the day-time pattern? Where is the wind coming from? If it is coming from the Atlantic I would assume it brings in radon poor sea air...

General comment on figures: Neither of these figures fits the current format: Unless the final article is two columns and wider, stack them on top of each other!

Section 3.3. GDR influencing by radon concentrations: I believe you can’t draw any meaningful conclusions here by simply looking at P and NP days. Rain events are typically very episodic and irregular, and not constant at all, particularly not over a full day. Rain events most of the time only last for a few hours at most, or vary a lot in intensity throughout rainy days. I suspect the differences between P and NP days are just statistical fluctuations: The differences between P and NP days are driven by a few rain events taking place at specific times (more events in the morning, vs more events in the afternoon). I would also suspect that the daily evolution of radon concentration changes with the seasons. So, there might have been more rain events during a specific season, where radon fluctuations are higher. All of this might affect the averages substantially and the conclusion P = more radon in the morning and less radon in the afternoon is just coincidental. The fact that the GDR doesn’t considerably change in P compared to NP supports my assumption. A better way of analyzing the data would be to actually correct the radon concentrations by their daily fluctuations. For example average 2-4 days (no precipitation) before and after a rainy day and subtract the average concentration from the radon concentration during the rain event (same for GDR). Then you align all rain events with the onset of rain, or look at them one by one. You should see a spike in radon during the rain (and even stronger GDR), that decays back to baseline over a few hours after the rain event. Doing so you don’t need to worry about P and NP, you will directly see what the impact of rain is.

Section 3.3: GDR evolution under GL with precipitation and non precipitation: study cases: First, the section number is wrong here, there already is a Section 3.3. While the section is well written and I generally agree with what is said, it is a bit surprising to me to say that an increase in radon from GL leads to more false alarms, while obviously a much larger increase of radon in the night does not have the same effect. I suspect there is some other effect, maybe the strong winds during GL affect both radon readings and GDR, but one doesn’t directly cause the other. I already mentioned this, but I also don’t think the conclusion of radon going down during a GL event is correct looking at Figure 11. There are clearly visible upwards bumps in radon whenever GDR goes up. They just sit on a larger trend of declining radon values, typical during the afternoon hours? I think this section needs to be considerably reworked. While those two examples are interesting, I think a similar analysis as I suggested for the previous Section would be appropriate. Subtract the average radon and GDR levels measured over a few nearby days from the GL event values. Then align all GL events when they start (onset) and you see the “average” GL evolution as a function of time. Then draw conclusions from this.

Conclusion (Line 334): I don’t understand this. Where do the authors show that GL and rain events are “accompanied by reductions in radon concentrations”? They show, at least to me, an “increase” in radon concentrations that slowly drops to baseline over the time of a few hours after the onset of the event, which is in line with literature.

Author contribution, Acknowledgments and conflict of interest section: They have not properly been completed!

Detailed comments:

Line 13: I would remove “automatic stations of”

Line 20: The present results point out THAT GDR values are OFTEN above the alarm levels when GL events are ON GOING? The current version of this phrase is not proper english.

Line 24: “The results demonstrate that GL events are accompanied by an increase (GL without precipitation) and a reduction (GL with precipitation) in radon concentrations, and that GL with precipitation are associated with higher increases in GDR”. This is a pretty strong statement that would go against current literature, which says precipitation increases radon concentrations. I don’t believe the paper shows this, even the radon levels in Figure 11 (the only place this is discussed) seem to suggest an increase during GL events (although small and on the flank of a falling radon curve from the daily radon cycle).

Line 64: Dot missing

Line 118: Set of double brackets, use — for example to go around this

Section 2.2: What meteorological data was collected? How was rainfall (precipitation) measured?

Line 124: I don't understand that sentence, maybe rephrase this as: "The Bilboa radiological station is a BAI9100 manufactured by Berthold". I don't understand why “trademark” would be important?

Line 125: What does the BAI9100 measure, what sensors are used, how does it calculate GDR?

Line 125:What is the "between other" doing here? Are you trying to say something along the lines of "it measures many things, but GDR is reported based on a proportional counter, and radon measurements are derived with the ABPD-Compensation and Measurement principle."

Line 135: “This level represents the value from which a signal measured by the instrument should be analyzed in detail to determine its origin”. This phrase doesn’t make any sense to me. I think it can just be removed.

Line 142: “a the” -> “the”

Line 146: No need to capitalize “Alarm Levels”. The paragraph talks about low and high confidence levels (“such as” meaning it is an example) but then doesn’t specify “the” alarm levels that were used on the system. How exactly is the alarm level set? Also a “confidence level” is the wrong word here, use threshold or simply alarm level. Confidence level describes how certain you are that something is happening or not, so for example if a signal is very high you are more confident it is not background. 

Line 153: I don’t understand this control. Does it say at least 75% of the duration of a given GL event needs to be covert by radiation data? Were there recent outages or why would an event not be covert? What happened in July that there is no GL event surviving the threshold? How many GL events were there in total before the cut?

Line 169: I don’t understand this formula for the mean standard deviation? I also never heard of the “mean standard deviation”, and I can’t really find much online either.

Line 173: No url here, add a proper reference at the end of the document.

Figure 4: So much wasted space between the bars. This should be more of a continuous distribution (histograms), and maybe use overlapping histograms for the second panel (I would need to see how it looks to decide if it is better), such as done here: https://i.pinimg.com/originals/41/67/03/416703cacdc80a61b25891e59c3efec4.jpg

Same for figure 5!

Line 206: what is “N”, do you mean “P”?

Line 207 - 210: Discuss averages or medians instead of “peak ranges” and “frequencies in the higher values and lower values”. I am pretty sure average(P) > average(NP) for all seasons. And yes probably we also have average(summer) < average (any other season). Maybe add them into a table?

Line 212: This is phrased a bit unfortunately. Precipations moves airborne radionuclides closer to the ground. Using remove here might raise questions why the rate increases in a GDR sensor, when it is gone. Our understanding is that radioisotopes are attached to droplets in the cloud and the rain brings it to the ground, leading to an increase in GDR. There is another, albeit less noticeable effect; the water on the ground creates a sort of blanket that prevents radon outgassing from the ground, which reduces the GDR slightly.

Line 214: I would have suspected that radon has a harder time moving from the ground to the atmosphere in winter (particular with snow), but I would assume that similar to rain, snow would trap some radioisotopes, but as it take a while for it to melt, retains them for longer. Not sure if there is a lot of snow in this area; It would be interesting to cross check if higher GDR’s are correlated with snow days?

Line 224: While I agree with this being likely true, I don’t know if it can easily be derived from Figure 6. It seems to be driven by two outlier events in February and March, while the rest seems to be pretty consistent with the average + spread in GDR. This is in line with my main concern with this article; It is hard to draw conclusions on ~hour long events by simply looking at daily averages!

Figure 6: I think it would be good to add “error bars” to this figure showing the standard deviations for the daily average or the range for the X’s percentile (with X=90% or whatever the authors deem valuable).

Line 227: I don’t see a single event in Figure 6 that has 0.0923 uSv/h? Maybe cross check this number.

Figure 7: I think it would be good to show the X percentile (X=90% or something) bands around each of these lines (with some level of transparency for better visibility).

There is no Figure 8: I think the numbering is off.

Line 248: Better title would be “Impact of radon concentrations on GDR”. “influencing” here is grammatically not correct.

Line 265: Again, while “removing from the atmosphere” is technically correct, I think it is a bad way of phrasing it for this context! They are TRANSPORTED with the rain from higher up in the atmosphere (mostly from the cloud forming regions) close to the surface.

Figure 9: Also here some sort of percentile bands around the central lines would be useful to better judge the variability.

Line 306: Figure number missing

Figure 11: Radon is green in the plot but gray in the legend. In general I think Figure 11 and Figure 10 are a bit overloaded. It would be good to have an axis for radon as well. You could add one more “radon” axis on the right, or just split them up into 3 separate plots with a common time axis. Also why does the alarm threshold increase during rain? The authors never talk about this mechanism.

317: temporarily reducing? I think you mean “increasing”? Again, it is only a displacement, not really a removal.

Comments on the Quality of English Language

See comments.

Author Response

Review 3: The authors discuss measurements observed in a network of sensors deployed in the North Eastern region of Spain, around the city of Balboa. In particular, they discuss how a weather event in the region, called Galerna (GL), typically characterized by strong winds, and precipitation affects readings of the Gamma Dose Rate (GDR) in their sensors. They divide their dataset into precipitation (P) and non precipitation (NP) data, as well as GL and non GL data and investigate differences between these segments. Supplementary they also investigate radon readings and look for correlations with the GDR. Finally they present two examples of GL events, one with precipitation and one without and show how the discussed effect, are realized in individual instances.

Overall, the authors present their work in a clear and simple-to-understand language, the paper is well structured, and they use proper English. Their discussion of existing literature and their overall results agrees with the understanding that I have of the topic. The presented work is interesting and is relevant to the readers of sensors. I do agree with the content of the paper and the overall conclusion, but I don’t believe that most of the data, as they are currently presented, actually support their claims. In order to study the impact of events that typically last a few hours and can occur at any time throughout the day, it is necessary to decouple the daily fluctuations from the event's impact. This has not been done here. I believe most observed effects, as presented, are statistical in nature and just happen to go the right way (and sometimes they also go the wrong way)! I believe that it is possible to correct the analysis (and figures) and find quantitative indicative results. I will provide suggestions on how to go about this for each of the relevant sections in the general comments section below. If for some reason the suggestions do not work out, or if the authors already tried this, they should talk about it in a “Discussions” section. To conclude, I strongly encourage the authors to address the flaws of the manuscript and continue pursuing a publication in this journal. I believe their work is interesting and having more results in this field of research is very valuable.

Many thanks for these words. We really appreciate these comments and for the following suggestions. We have implemented all of them in the new version of this article.

One final comment: There are a lot of small errors (wrong Figure and Section numbers, for example) in the script. A proper proofreading should have caught those. It is not a reviewer's job to proofread a paper, the authors should have done so themself.

We deeply regret it. We have tried to solve all of them in the new version of this manuscript

General comments:

Section 2.3: I don’t think it is entirely clear that you have two alarm levels. One being active whenever there is rain, another (lower one) whenever there is no rain. Try to make this a bit clearer

We have modified the new version of the article to clarify this point. We have included the following information Lines 134-142 “It is essential to identify and understand the increase in GDR values. Using the recorded GDR values, two distinct alert thresholds are defined based on weather conditions: one for dry conditions and another for wet conditions. Due to the washout of the radon daughter, which are located in the soil when there is precipitation, the threshold of this situation has to be higher than the other. The GDR values for both scenarios follow a normal distribution, and given that the data consist of discrete values, we apply Currie's definition [12] to establish critical levels of GDR. The alert threshold for dry weather is activated in the absence of precipitation, while the wet weather threshold becomes active during precipitation events. These thresholds adjust dynamically depending on the presence or absence of rainfall, ensuring that variations in GDR are properly assessed according to prevailing weather conditions.”

Also, I think the discussion of the alarm level here is cryptic. Maybe just give some examples and provide the false alarm rate. At k=1 you will have about 15.8% of backgrounds triggering an alarm (or roughly one false alarm every hour), at k=2 you have about 2.3% of backgrounds doing the same (or roughly a false alarm every 7 hours), k=3 about 0.1% of backgrounds (false alarm rate of 1 every 5 days). Please check my math. I personally would just state the false alarm rate for a few examples of k.

Thanks for your comment. We know that in other Networks the Alarm Levels are defined by static level and with K=3, but in some cases, small increases of GRD are not detected. For example, close to our Bilbao station there is an incinerator and in one case iodine plume was detected only with the increase of GDR in dry case. We could change the value of K, but we are comfortable with those levels.

Section 2.4: What is the average length of a GL event? Would be good to mention this. Is it a few minutes, hours, or days? Is the GL event starting randomly or is there a correlation with the day-time pattern? Where is the wind coming from? If it is coming from the Atlantic I would assume it brings in radon poor sea air...

We understand this comment and the suggestion. We have included more information about GL in the new version of this paper Lines 165-169 “[9] characterized the occurrence and development of this event, indicating that it occurs more frequently between May and June, and between noon and the late afternoon. More information about GL can be consulted in [9].”

General comment on figures: Neither of these figures fits the current format: Unless the final article is two columns and wider, stack them on top of each other!

We have modified the figure accordingly.

Section 3.3. GDR influencing by radon concentrations: I believe you can’t draw any meaningful conclusions here by simply looking at P and NP days. Rain events are typically very episodic and irregular, and not constant at all, particularly not over a full day. Rain events most of the time only last for a few hours at most, or vary a lot in intensity throughout rainy days. I suspect the differences between P and NP days are just statistical fluctuations: The differences between P and NP days are driven by a few rain events taking place at specific times (more events in the morning, vs more events in the afternoon). I would also suspect that the daily evolution of radon concentration changes with the seasons. So, there might have been more rain events during a specific season, where radon fluctuations are higher. All of this might affect the averages substantially and the conclusion P = more radon in the morning and less radon in the afternoon is just coincidental. The fact that the GDR doesn’t considerably change in P compared to NP supports my assumption. A better way of analyzing the data would be to actually correct the radon concentrations by their daily fluctuations. For example average 2-4 days (no precipitation) before and after a rainy day and subtract the average concentration from the radon concentration during the rain event (same for GDR). Then you align all rain events with the onset of rain, or look at them one by one. You should see a spike in radon during the rain (and even stronger GDR), that decays back to baseline over a few hours after the rain event. Doing so you don’t need to worry about P and NP, you will directly see what the impact of rain is.

Thank you very much for this suggestion. In order to establish the impact and difference between days with and without GL we have included Figure 6 in the new version of this article:

We have included the following information in the new version of this manuscript Lines 253-263 “In this section, 40 days of GL covering the period 2009 to 2018 are used as a reference, having all of them sufficient hourly GDR data available for analysis. Figure 6 shows the overall average daily GDR cycle in Bilbao without GL (NGL) and the GDR cycle under GL events (GL). On average, GDR values under GL are higher throughout the entire day. It is notable that both cycles exhibit high values during the night, followed by a decline to lower values. However, from midday onwards, GDR concentrations rise under GL events, with an increase from 12:00 hours until 20:00 hours, after which they begin to decrease. This increase coincides with the period established in [9] for the development of GL. Therefore, the GDR cycles in Bilbao reveals distinct daily patterns, suggesting a positive influence of GL formation in GDR levels during the afternoon, emphasizing the impact of atmospheric conditions on radiation levels throughout the day.”

In addition, we have modified Figure 8 to investigate the impact of precipitation under GL events. We have applied the following methodology: Instead of considering days before and after the GL event, we have taken as reference the GDR daily cycle under GL event, and we have calculated the difference between this cycle and the two obtained with precipitation and without precipitation respectively.

We have included the following information in the new version of this article Lines 280-293 “Figure 8 displays the hourly evolution of GDR under GL events, considering pre-cipitation and without precipitation. Both daily cycles are compared with the overall average of GDR under GL events for the period 2009-2018. The figure shows that on GL with precipitation (P), daily GDR concentrations are higher than those on GL days without precipitation (NP). Similar values for P and NP events are registered at night and in the morning. While those obtained for NP days present similar values to the overall average, GDR for P days are slightly higher. Notably, after 12:00 hours, coinciding with the period in which [9] indicates the onset of GL development, the GDR on P days (red line) consistently exceeds both NP days (blue line) and the overall average (black line). This indicates a significant increase in the afternoon in GDR levels during GL events under precipitation, suggesting that precipitation may enhance the washout of radionuclides from the atmosphere, leading to elevated GDR levels. This evolution highlights that precipitation amplifies the effects of GL on GDR concentrations, particularly during critical periods of GL formation”

These figures clearly demonstrated that GL occur around midday or early afternoon, we considered that it was not necessary to establish a common start time for all episodes.

We have also performed the same analysis for radon concentrations, and we have compared GDR and radon differences under GL with precipitation and without precipitation. The following figure has been included in the new version of the paper. This figure helps to understand the relationship between both, GDR and radon, under GL with precipitation and without precipitation

Section 3.3: GDR evolution under GL with precipitation and non precipitation: study cases: First, the section number is wrong here, there already is a Section 3.3.

It has been changed

While the section is well written and I generally agree with what is said, it is a bit surprising to me to say that an increase in radon from GL leads to more false alarms, while obviously a much larger increase of radon in the night does not have the same effect. I suspect there is some other effect, maybe the strong winds during GL affect both radon readings and GDR, but one doesn’t directly cause the other. I already mentioned this, but I also don’t think the conclusion of radon going down during a GL event is correct looking at Figure 11. There are clearly visible upwards bumps in radon whenever GDR goes up. They just sit on a larger trend of declining radon values, typical during the afternoon hours? I think this section needs to be considerably reworked. While those two examples are interesting, I think a similar analysis as I suggested for the previous Section would be appropriate. Subtract the average radon and GDR levels measured over a few nearby days from the GL event values. Then align all GL events when they start (onset) and you see the “average” GL evolution as a function of time. Then draw conclusions from this.

To understand the relationship between both, GDR and radon, we have included the previous figure, which helps to understand the impact of radon concentrations on GDR. In section 3.4 we provide two examples of GL without precipitation and with precipitation. You are right, and there was a mistake in the way to describe the evolution of radon and GDR in the case of non precipitation. We have included the following information in the new version of this article (Lines 351-357) “During this increase in radon concentrations, GDR values remains in similar values, and only a very small increase is registered, reaching some of the GDR values the alarm level, and therefore, these occurrences should thus be classified as false alarms. However, they cannot be only associated with an increase in radon concentrations. The sudden increase in surface winds during the GL [9] can affect both radon and GDR concentrations, so, it is not possible to establish a direct relationship between them in this specific case.”

Conclusion (Line 334): I don’t understand this. Where do the authors show that GL and rain events are “accompanied by reductions in radon concentrations”? They show, at least to me, an “increase” in radon concentrations that slowly drops to baseline over the time of a few hours after the onset of the event, which is in line with literature.

Yes, you are right, and we have deleted this information in the new version of this article.

Author contribution, Acknowledgments and conflict of interest section: They have not properly been completed

You are right. It has been included in the new version of the manuscript.

Detailed comments:

Line 13: I would remove “automatic stations of”

It has been removed in the new version of this manuscript

Line 20: The present results point out THAT GDR values are OFTEN above the alarm levels when GL events are ON GOING? The current version of this phrase is not proper english.

We have removed this sentence and we have included the following one Line 21 “The present results indicate that GDR values frequently exceed alarm levels when GL are formed”

Line 24: “The results demonstrate that GL events are accompanied by an increase (GL without precipitation) and a reduction (GL with precipitation) in radon concentrations, and that GL with precipitation are associated with higher increases in GDR”. This is a pretty strong statement that would go against current literature, which says precipitation increases radon concentrations. I don’t believe the paper shows this, even the radon levels in Figure 11 (the only place this is discussed) seem to suggest an increase during GL events (although small and on the flank of a falling radon curve from the daily radon cycle).

We have modified the information provided. Lines 24-27 “The results demonstrate that GL event with precipitation is a meteorological scenario that can be associated with false alarm. The present analysis provides a distinct contrast in radon behavior under the same meteorological event in case of being developed with precipitation or without precipitation.”

Line 64: Dot missing

It has been added in the new version of this manuscript.

Line 118: Set of double brackets, use — for example to go around this.

It has been modified in the new version of this manuscript “observations, such as temperature  (◦C) and precipitation (mm)”

Section 2.2: What meteorological data was collected? Temperature, Relative humidity, atmospheric pressure, solar irradiation, wind direction and speed and rain. How was rainfall (precipitation) measured?

It has been modified in the new version of this manuscript “precipitation (mm)”

Line 124: I don't understand that sentence, maybe rephrase this as: "The Bilbao radiological station is a BAI9100 manufactured by Berthold". I don't understand why “trademark” would be important?

You are totally right. This information has been deleted in the new version of this manuscript

Line 125: What does the BAI9100 measure, what sensors are used, how does it calculate GDR? Alpha concentration, beta concentration, radon, iodine, GDR. For alpha, beta and radon ZnS plastic scintillator, for iodine NaI detector, and for GDR Proportional Counter.

We have modified the information about BAI9100 in the new version of the manuscript “The Bilbao radiological station is equipped with a BAI9100, which measures alpha and beta particle concentrations, iodine, radon, and gamma dose rate (GDR). Alpha, beta, and radon are detected using a ZnS plastic scintillator, iodine is measured with a NaI detector, and GDR is monitored with a proportional counter”

Line 125: What is the "between other" doing here? Are you trying to say something along the lines of "it measures many things, but GDR is reported based on a proportional counter, and radon measurements are derived with the ABPD-Compensation and Measurement principle."

We have modified this information in the new version of the manuscript “The Bilbao radiological station is equipped with a BAI9100, which measures alpha and beta particle concentrations, iodine, radon, and gamma dose rate (GDR). Alpha, beta, and radon are detected using a ZnS plastic scintillator, iodine is measured with a NaI detector, and GDR is monitored with a proportional counter.”

Line 135: “This level represents the value from which a signal measured by the instrument should be analyzed in detail to determine its origin”. This phrase doesn’t make any sense to me. I think it can just be removed.

It has been removed in the new version of this manuscript

Line 142: “a the” -> “the”

It has been changed in the new version of this manuscript

Line 146: No need to capitalize “Alarm Levels”.

It has been changed in the new version of the article

Line 146 The paragraph talks about low and high confidence levels (“such as” meaning it is an example) but then doesn’t specify “the” alarm levels that were used on the system. How exactly is the alarm level set? Also a “confidence level” is the wrong word here, use threshold or simply alarm level. Confidence level describes how certain you are that something is happening or not, so for example if a signal is very high you are more confident it is not background.

We understand your doubts about this information. We have modified this information in the new version of this manuscript “The procedure is the following: Each time that a GDR measurement is recorded, it is checked whether there is a record of precipitation or not. Based on this, the recorded value is compared to the corresponding alarm level. If the GDR value exceeds the alarm level, an automatic notification is triggered. Proportional counters are affected by storms, fog, voltage fluctuations, nearby construction activities, and other factors. For this reason, this procedure must be well-defined, as setting the alarm threshold too high could allow critical alerts to be missed, while setting it too low would trigger continuous alerts, diminishing their significance. Below, you can find the table with the alarm levels defined from 2009 to 2019 for wet and dry conditions. We include this table in article because we think it is relevant.

YEAR

Dry conditions (uSv/h)

Wet conditions(uSv/h)

2009

0,0841

0,0941

2010

0,0839

0,0930

2011

0,0830

0,0920

2012

0,0825

0,0914

2013

0,0828

0,0913

2014

0,0815

0,0900

2015

0,0839

0,0949

2016

0,0838

0,0913

2017

0,0829

0,0927

2018

0,0825

0,0904

 Line 153: I don’t understand this control. Does it say at least 75% of the duration of a given GL event needs to be covert by radiation data? Were there recent outages or why would an event not be covert? What happened in July that there is no GL event surviving the threshold? How many GL events were there in total before the cut?

Many thanks for this question

  • Yes, there is a need to have 75 % of 10-min GDR data to consider a GL event in the present analysis. Therefore, GL events with less than 75 % of the data available, are not considered in the analysis.
  • In July there were GL events with percentage of data available less than 75 %, so they were discarded in the present analysis

Line 169: I don’t understand this formula for the mean standard deviation? I also never heard of the “mean standard deviation”, and I can’t really find much online either.

Sorry, there is a mistake. We have modified this information in the new version of this manuscript “We have calculated the standard error of the mean, which is calculated Sx/N1/2 This value provides a measure of how precisely the sample mean estimates the true population mean. Unlike the standard deviation, which measures the dispersion of individual values around the mean, the standard error reflects the uncertainty in the estimate of the mean.

Line 173: No url here, add a proper reference at the end of the document

We have removed it in the new version of the document.

Figure 4: So much wasted space between the bars. This should be more of a continuous distribution (histograms), and maybe use overlapping histograms for the second panel (I would need to see how it looks to decide if it is better), such as done here: https://i.pinimg.com/originals/41/67/03/416703cacdc80a61b25891e59c3efec4.jpg

Same for figure 5!

Sorry but we do not agree with this suggestion, we consider that this way of presenting these results is appropriated.

Line 206: what is “N”, do you mean “P”?

We regret this mistake. You are right.

Line 207 - 210: Discuss averages or medians instead of “peak ranges” and “frequencies in the higher values and lower values”. I am pretty sure average(P) > average(NP) for all seasons. And yes probably we also have average(summer) < average (any other season). Maybe add them into a table?

We have included this information in the new version of this article: ”The seasonal GDR values with and without precipitation show higher concentrations during precipitation events across all seasons. In both meteorological scenarios, the highest average values are reached in winter (0.0769 ± 0,0002 µSv and 0.0765 ± 0,0001 µSv/h, for precipitation and without precipitation, respectively), while the lowest values are observed in summer (0.0754 ± 0,0002 µSv and 0.0751 ± 0,0001 µSv/h, respectively)”

Line 212: This is phrased a bit unfortunately. Precipitations moves airborne radionuclides closer to the ground. Using remove here might raise questions why the rate increases in a GDR sensor, when it is gone. Our understanding is that radioisotopes are attached to droplets in the cloud and the rain brings it to the ground, leading to an increase in GDR. There is another, albeit less noticeable effect; the water on the ground creates a sort of blanket that prevents radon outgassing from the ground, which reduces the GDR slightly.

We have modified the sentence in the new version of this article: Line XX: “These results demonstrate the significant influence of precipitation on GDR concentrations, primarily due to the washout effect, where rain brings airborne radionuclides closer to the ground, leading to increased GDR levels during and shortly after rainfall events. Additionally, the accumulation of water on the ground can slightly reduce radon outgassing, tempering the increase in GDR.”

Line 214: I would have suspected that radon has a harder time moving from the ground to the atmosphere in winter (particular with snow), but I would assume that similar to rain, snow would trap some radioisotopes, but as it take a while for it to melt, retains them for longer. Not sure if there is a lot of snow in this area; It would be interesting to cross check if higher GDR’s are correlated with snow days?

Many thanks for this suggestion. We understand this question, and it would be great to carry out this comparison. However, we do not have data of snow days. It can be an interesting analysis in the future.

Line 224: While I agree with this being likely true, I don’t know if it can easily be derived from Figure 6. It seems to be driven by two outlier events in February and March, while the rest seems to be pretty consistent with the average + spread in GDR. This is in line with my main concern with this article; It is hard to draw conclusions on ~hour long events by simply looking at daily averages!

 We understand the doubts. For this reason we have include the standard error of the mean in each figure.

Figure 6: I think it would be good to add “error bars” to this figure showing the standard deviations for the daily average or the range for the X’s percentile (with X=90% or whatever the authors deem valuable).

 We understand the doubts. For this reason we have include the standard error of the mean in each figure.

Line 227: I don’t see a single event in Figure 6 that has 0.0923 uSv/h? Maybe cross check this number.

Yes, there was a mistake.

Figure 7: I think it would be good to show the X percentile (X=90% or something) bands around each of these lines (with some level of transparency for better visibility).

We understand the doubts. For this reason we have include the standard error of the mean in each figure.

There is no Figure 8: I think the numbering is off.

You are right, and we regret the mistake. The number of the figures has been changed accordingly.

Line 248: Better title would be “Impact of radon concentrations on GDR”. “influencing” here is grammatically not correct.

It has been changed in the new version of this manuscript

Line 265: Again, while “removing from the atmosphere” is technically correct, I think it is a bad way of phrasing it for this context! They are TRANSPORTED with the rain from higher up in the atmosphere (mostly from the cloud forming regions) close to the surface.

We agree with you, and it has been changed in the new version of this paper.

Figure 9: Also here some sort of percentile bands around the central lines would be useful to better judge the variability.

We understand the doubts. For this reason we have include the standard error of the mean in each figure.

Line 306: Figure number missing

It has been changed in the new version of this article

Figure 11: Radon is green in the plot but gray in the legend. In general, I think Figure 11 and Figure 10 are a bit overloaded. It would be good to have an axis for radon as well. You could add one more “radon” axis on the right, or just split them up into 3 separate plots with a common time axis.

Also why does the alarm threshold increase during rain? The authors never talk about this mechanism.

Information about the alarm threshold has been included in the new version of this manuscript.

317: temporarily reducing? I think you mean “increasing”? Again, it is only a displacement, not really a removal.

It has been changed in the new version of this manuscript

Round 2

Reviewer 3 Report

Comments and Suggestions for Authors

I think this is a clear improvement over the last version and the authors addressed most if not all of my comments. I have one remaining major issue, related to the anti correlation between GRD measurements and radon concentrations during a rain event. I am just not capable of following the author's argument and I presented my own attempt of interpretation in the detailed comments section (I am not sure it has any legimacy though). I think a more consistent discussion is required, thus I mark this as a minor revision. All the other points are mostly grammatical in nature and I believe the authors should be able to adjust their manuscript accordingly in little time.

Line 48-51: Run on sentence, my suggestion “... granitic regions. Another study …” and “... impact on GDR and noted higher … “

Line 82: First 3) -> 2)

Line 120: degree Celcius symbol looks odd (circle should be superscript)

Line 134: I feel the first sentence misses something. I would suggest adding to the beginning of the paragraph the following: “To define alarm levels, it is essential to identify and understand fluctuations in GDR values.”

Line 169: not GL -> no GL.

Line 172: I am still not sure what a threshold of 75% means. I think it would be necessary to mention that the stations sometimes don’t record data or that there is some sort of down time. Are the authors not comfortable with the fact that the device doesn’t record data for some period of time? I just don’t understand what is wrong with being blunt about it if it is the case. If it is not, then I don’t understand why not all GL events have 100% coverage. Some background would be useful, I can only guess.

Line 173: not GL -> no GL

Line189: he -> the

Line: 188-193: I think the definition of the standard error and its meaning is known to the reader and doesn’t need to be repeated, “which is calculated by … with larger samples” can be removed (I am fine having it though). I was confused about the term that was used previously, that is why I commented on it, not because I thought it lacks clarity here.

Line 253-254: “In this section, 40 days of GL covering the period 2009 to 2018 are used as a reference having all of them sufficient hourly GDR data available for analysis.” -> “In this section, 40 days with GL events, covering the period 2009 to 2018, are used. They all have sufficient hourly GDR data available for analysis.” The current version is grammatically incorrect.

Line 254-263 + Figure 6: This is a very nice discussion now. I agree with the findings and consider this very valuable information. It is interesting that mornings have statistically significant increased levels of GDR during a GL event, which warrants the question of if GDR levels actually are predictive of GL events. I don’t know if the data is sufficient to answer this question, but it is certainly interesting to see it.

Line 270: I personally would remove the “it is calculated that”.

Line 280: “considering” -> “separating events with”

Line 282: “that on GL”, “that during GL events”

Line 283: “those on GL days” -> “those during GL days”.

Line 280 - 293 + Figure 8: This now is also a very nice discussion of the data. And I agree that the data now support the conclusion the authors draw. 

Line 298-309: and Figure 9: First the authors have a wrong figure number in line 307 and 311 (it should be Figure 9a and Figure 9b). Next, I feel the new figure 9 is much harder to read than the previous one. I think what the authors should do instead, is to use the first version of the plot, but add the cycle average to each plot, so that both, the daily pattern, but also the excess and reduction from the average is clearly visible. To be honest it would be best to actually show the average P and NP radon evolution from all the non GL days as well or instead. Currently the facts described in the text (Line 300 - 304) is not visually presented to the reader at all. I very much agree with the remaining discussion in this section (Line 309 - 314) and think it is very interesting to see the anti-correlation (P: lower radon, but higher GDR, NP: higher radon but lower GDR).

Line 316 - 330: The authors do not discuss the anti correlation between radon concentration and GDR at all. During a P event radon is moved near the ground, which increases GDR. That makes sense and is mentioned. And yes during a NP GL radon is "blown away" and thus GDR is lower. But why does more radon not mean higher radon concentrations, and less radon not lower radon concentration? All I can think of is the different modalities of measurement. Radon might be measured towards the top of the station, I believe the way the radon system works is that air is sucked in from the outside and flows over a ZnS plastic scintillator counting alpha and beta particles. Gamma rays might be measured more towards the bottom, from all directions homogeneously. That would mean the radon counter only sees the radon in the air, which indeed can be reduced during a rain event, because all the radon was “forced” to be confined really close to the ground, on the ground itself, or even below the ground (and increased during NP due to strong winds, blowing it up from the ground or nearby areas). The gamma’s from ground bound radon can escape and still hit the proportional counter increasing the GDR during rain (and obviously less radon in, or near the ground would mean less GDR during a NP GL event). It could also be that the two effects are not related at all. Maybe radon decreases during a GL rain event (and increases during a NP GL event), but the increase (decrease) in GDR is decoupled and comes from elsewhere? I don’t know if the authors can work around this issue without having to make any statements. It honestly is the weakest point of the publication at this point. It would be okay to discuss some explanations and say that they simply don’t understand it either to be honest, but it needs to be consistently addressed.

Line 351-357: A little clunky: Suggestion “During this increase in radon concentrations, GDR values remain mostly constant, but over time increase slightly; Some GDR values eventually reach the alarm level. These occurrences should be classified as false alarms. However, it is not possible to establish a direct correlation of these false alarms with radon concentration, as the sudden increase in surface winds during the GL [9] affect both rand and GDR concentrations independently.”

Comments on the Quality of English Language

All comments are in the general section.

Author Response

I think this is a clear improvement over the last version and the authors addressed most if not all of my comments. I have one remaining major issue, related to the anti correlation between GRD measurements and radon concentrations during a rain event. I am just not capable of following the author's argument and I presented my own attempt of interpretation in the detailed comments section (I am not sure it has any legimacy though). I think a more consistent discussion is required, thus I mark this as a minor revision. All the other points are mostly grammatical in nature and I believe the authors should be able to adjust their manuscript accordingly in little time.

Line 48-51: Run on sentence, my suggestion “... granitic regions. Another study …” and “... impact on GDR and noted higher … “

It has been modified in the new version of this paper (Lines 49-52)

Line 82: First 3) -> 2)

It has been modified in the new version of this paper (Lines 84)

Line 120: degree Celcius symbol looks odd (circle should be superscript)

It has been modified in the new version of this paper (Lines 122)

Line 134: I feel the first sentence misses something. I would suggest adding to the beginning of the paragraph the following: “To define alarm levels, it is essential to identify and understand fluctuations in GDR values.”

It has been modified in the new version of this paper (Lines 134-135)

Line 169: not GL -> no GL.

It has been modified in the new version of this paper (Lines 172)

Line 172: I am still not sure what a threshold of 75% means. I think it would be necessary to mention that the stations sometimes don’t record data or that there is some sort of down time. Are the authors not comfortable with the fact that the device doesn’t record data for some period of time? I just don’t understand what is wrong with being blunt about it if it is the case. If it is not, then I don’t understand why not all GL events have 100% coverage. Some background would be useful, I can only guess.

We agree with you. It has been modified in the new version of this paper (Lines 171-174)

Line 173: not GL -> no GL

It has been modified in the new version of this paper (Lines 175)

Line189: he -> the

It has been modified in the new version of this paper (Lines 189)

Line: 188-193: I think the definition of the standard error and its meaning is known to the reader and doesn’t need to be repeated, “which is calculated by … with larger samples” can be removed (I am fine having it though). I was confused about the term that was used previously, that is why I commented on it, not because I thought it lacks clarity here.

It has been removed in the new version of this paper (Lines 189-190)

Line 253-254: “In this section, 40 days of GL covering the period 2009 to 2018 are used as a reference having all of them sufficient hourly GDR data available for analysis.” -> “In this section, 40 days with GL events, covering the period 2009 to 2018, are used. They all have sufficient hourly GDR data available for analysis.” The current version is grammatically incorrect.

It has been modified in the new version of this paper (Lines 253-254)

Line 254-263 + Figure 6: This is a very nice discussion now. I agree with the findings and consider this very valuable information. It is interesting that mornings have statistically significant increased levels of GDR during a GL event, which warrants the question of if GDR levels actually are predictive of GL events. I don’t know if the data is sufficient to answer this question, but it is certainly interesting to see it.

Many thanks for this comment

Line 270: I personally would remove the “it is calculated that”.

It has been removed in the new version of this paper (Line 270)

Line 280: “considering” -> “separating events with”

It has been modified in the new version of this paper (Lines 280-281)

Line 282: “that on GL”, “that during GL events”

It has been modified in the new version of this paper (Lines 282)

Line 283: “those on GL days” -> “those during GL days”.

It has been modified in the new version of this paper (Lines 283-284)

Line 280 - 293 + Figure 8: This now is also a very nice discussion of the data. And I agree that the data now support the conclusion the authors draw. 

Many thanks for this comment.

Line 298-309: and Figure 9: First the authors have a wrong figure number in line 307 and 311 (it should be Figure 9a and Figure 9b). Next, I feel the new figure 9 is much harder to read than the previous one. I think what the authors should do instead, is to use the first version of the plot, but add the cycle average to each plot, so that both, the daily pattern, but also the excess and reduction from the average is clearly visible. To be honest it would be best to actually show the average P and NP radon evolution from all the non GL days as well or instead. Currently the facts described in the text (Line 300 - 304) is not visually presented to the reader at all. I very much agree with the remaining discussion in this section (Line 309 - 314) and think it is very interesting to see the anti-correlation (P: lower radon, but higher GDR, NP: higher radon but lower GDR).

We have modified Figure 9 in the new version of this paper. We have included the following plot dealing with radon concentrations. Considering that Figure 8 presents daily cycles of GDR under GL events with precipitation (GL-P), non precipitation (GL-NP) and the overall average during the period 2009-2018, and Figure 9 shows the radon results, the discussion included in section 3 is written based on both figures (Lines 331-341).

Figure 9. Daily cycles of radon concentrations under GL events with precipitation (GL-P), non precipitation (GL-NP) and the overall average during the period 2009-2018.

Line 316 - 330: The authors do not discuss the anti correlation between radon concentration and GDR at all. During a P event radon is moved near the ground, which increases GDR. That makes sense and is mentioned. And yes during a NP GL radon is "blown away" and thus GDR is lower. But why does more radon not mean higher radon concentrations, and less radon not lower radon concentration? All I can think of is the different modalities of measurement. Radon might be measured towards the top of the station, I believe the way the radon system works is that air is sucked in from the outside and flows over a ZnS plastic scintillator counting alpha and beta particles. Gamma rays might be measured more towards the bottom, from all directions homogeneously. That would mean the radon counter only sees the radon in the air, which indeed can be reduced during a rain event, because all the radon was “forced” to be confined really close to the ground, on the ground itself, or even below the ground (and increased during NP due to strong winds, blowing it up from the ground or nearby areas). The gamma’s from ground bound radon can escape and still hit the proportional counter increasing the GDR during rain (and obviously less radon in, or near the ground would mean less GDR during a NP GL event). It could also be that the two effects are not related at all. Maybe radon decreases during a GL rain event (and increases during a NP GL event), but the increase (decrease) in GDR is decoupled and comes from elsewhere? I don’t know if the authors can work around this issue without having to make any statements. It honestly is the weakest point of the publication at this point. It would be okay to discuss some explanations and say that they simply don’t understand it either to be honest, but it needs to be consistently addressed.

Yes, we agree with you. You are totally right. The way the radon system works is that air is drawn in from outside and flows over a ZnS plastic scintillator that counts alpha and beta particles. Gamma rays might be measured more consistently at the bottom, from all directions, in a homogeneous manner. This suggests that the radon counter detects only the radon in the air, which can indeed decrease during rain events because the radon is "forced" to remain very close to the ground, on the ground itself, or even below it. During normal conditions (NP), radon follows a daily cycle of emanation, which can be altered by precipitation. Natalia Alegría's thesis has already studied and successfully explained some of the increases in GDR as a function of radon and precipitation, and we are completing another study on the correlation between radon and GDR as a function of cloud height and washout, which we hope to publish in the future. Following your recommendation, we have included in the new version of this paper the following information:” We can also include in the analysis of the relationship between GDR and radon the different modalities of measurements. The way the radon system works is that air is drawn in from outside and flows over a ZnS plastic scintillator that counts alpha and beta particles. Gamma rays might be measured more consistently at the bottom, from all di-rections, in a homogeneous manner. This suggests that the radon counter detects only the radon in the air, which can indeed decrease during rain events because the radon is "forced" to remain very close to the ground, on the ground itself, or even below it. In this sense, during normal conditions (NP), radon follows a daily cycle of emanation, which can be altered by precipitation. Alegría et al. [22], it presents the impact of different factors, such as the measure methodology in the correlation between GDR and radon concentration.”

Line 351-357: A little clunky: Suggestion “During this increase in radon concentrations, GDR values remain mostly constant, but over time increase slightly; Some GDR values eventually reach the alarm level. These occurrences should be classified as false alarms. However, it is not possible to establish a direct correlation of these false alarms with radon concentration, as the sudden increase in surface winds during the GL [9] affect both rand and GDR concentrations independently.”

It has been modified in the new version of this paper (Lines 360-365)
